# Sericin from Fibroin-Deficient Silkworms Served as a Promising Resource for Biomedicine

**DOI:** 10.3390/polym15132941

**Published:** 2023-07-04

**Authors:** Yurong Li, Yongkang Wei, Guozheng Zhang, Yeshun Zhang

**Affiliations:** 1College of Biotechnology, Jiangsu University of Science and Technology, Zhenjiang 212100, China; yrli_0818@163.com (Y.L.); wwweiyk@163.com (Y.W.); zgzsri@163.com (G.Z.); 2Key Laboratory of Silkworm and Mulberry Genetic Improvement, Ministry of Agriculture and Rural Affairs, Sericultural Research Institute, Chinese Academy of Agricultural Sciences, Zhenjiang 212100, China; 3Zhenjiang Zhongnong Biotechnology Co., Ltd., Zhenjiang 212121, China

**Keywords:** sericin, fibroin-deficient silkworm, biomaterials, tissue engineering, drug delivery, cosmetics

## Abstract

Sericin, a fascinating natural biomaterial derived from silkworms, has received increasing interest in recent years for its unique bioactivity and high compatibility. Silkworms can be divided into wild-type or silk fibroin-deficient mutants according to whether they synthesize and secrete silk fibroin. Silk fibroin-deficient mutant silkworms and their cocoons are convenient for us to obtain diverse and high-quality sericin, which has been applicated in various fields such as cell culture, tissue engineering, drug delivery, and cosmetics. Here, we present an overview of our silkworm varieties resources, especially silk fibroin-deficient mutant silkworms. We optimized various extraction methods of sericin and summarized the characteristics and advantages of sericin. Finally, we developed and discussed a series of sericin-based biomaterials for promising applications for a diverse set of needs.

## 1. Introduction

The silkworm, as a completely domesticated insect with a good breeding and genetic system, has been considered an ideal model organism for biological study [1]. Silkworms can be divided into wild-type or silk fibroin-deficient mutants according to whether they synthesize and secrete silk fibroin. The wild-type silkworms secrete natural silk composed of sericin and silk fibroin. The sericin, as the outer layer, tightly encapsulates two bundles of silk fibroin (the core layer) with rounded triangle cross sections to protect and glue, increasing the stability and toughness of the silk [2,3]. Silk fibroin has attracted much attention in biomedical fields due to its suitable physicochemical properties and good biocompatibility [4]. In contrast, sericin has been neglected for thousands of years during the development of the textile industry, resulting in environmental pollution and a waste of resources [5,6]. In recent years, much research has found that sericin is an excellent natural biological material because of its excellent biological activity, low immunogenicity, controlled degradability, and easy modifiability [7,8,9]. However, separating sericin from wild-type silkworm cocoons seriously damages its natural structure and activity, accompanied with a complicated extraction process, thus restricting its application [10,11]. In addition, the content of sericin in wild-type silkworm cocoons is relatively low, accounting for about 25–30% [12,13]. Therefore, there is an urgent need to cultivate fine breeds of silk fibroin-deficient mutants, which only secrete pure sericin with high quality and not silk fibroin, due to the absence of posterior silk glands that synthesize silk fibroin. Our lab members obtained various silk fibroin-deficient mutant silkworms through genetic breeding technology [14,15]. Through establishing or optimizing various extraction methods of sericin from silk fibroin-deficient mutants, the properties of sericin were explored, including promoting cell adhesion and proliferation, humidity preservation, antibacterial, anti-oxidation, and anti-inflammatory properties, the inhibition of tyrosinase, etc. [16,17]. Moreover, sericin-based biomaterials with excellent properties (such as elasticity, mechanical strength, biodegradability, and biocompatibility) have been developed by modification, or by cross-linking with other polymers [7,18,19]. Therefore, sericin has attracted more and more attention in various fields, such as cell culture, tissue engineering, and drug delivery [16,17,20,21,22,23]. For instance, sericin has been used as a fetal bovine serum (FBS) substitute for cell cultures, promoting cell growth and proliferation [23]. The applications of sericin in tissue engineering involve tissue injury repair and the regeneration of bone and nerve [21,24,25,26]. Sericin-based biomaterials, such as hydrogel, film, and sponge, are usually used as carriers for delivering drugs or growth factors [27,28,29]. Sericin from silk fibroin-deficient silkworms was also applied in disease therapy, mainly referring to chronic nerve compression, myocardial infarction, and stroke [20,30,31]. Sericin has attracted great attention in the field of cosmetics due to its natural biological activities [32,33]. The recovery and exploitation of sericin would not only minimize environmental problems, but also has important scientific significance and application values. This review focused on concluding the resources and characteristics of sericin from silk fibroin-deficient mutants, summarizing different types of sericin-based biomaterials and their applications, and offering prospects for future applications.

## 2. Resources of Sericin and Cultivation of Silkworm Varieties

Sericin, a natural protein secreted by silkworms, consists of 18 amino acids [7]. Sericin has high yields (annual output of about 50,000 tons worldwide [34]), benefiting from silkworm varieties’ rich resources. There are thousands of silkworm varieties in the world, and our institute (Sericultural Research Institute, Chinese Academy of Agricultural Sciences) has more than a thousand species. The variety of silkworms provide us with different properties of sericin.

Cocoons produced by silk fibroin-deficient mutant silkworms are favored for separating and extracting sericin with excellent properties. However, considering that the natural mutant silkworms have low yields of sericin, poor resistance against disease, and breeding difficulty [14], our research lab focused on breeding new silk fibroin-deficient mutants to overcome the above defects. We obtained silk fibroin-deficient mutants with high yield, disease resistance, large cocoon, and small pupa by using silk fibroin-deficient mutant varieties, Nuclear Polyhedrosis Virus (NPV) resistance varieties, and high-yield and high-quality practical varieties as parents through silkworm genetic breeding techniques (cross and backcross) for dozens of generations. The sericin cocoon layer ratio produced by new mutant silkworms was increased from about 1.0% to 5.5%. These superior varieties have laid a solid foundation for exploiting and utilizing sericin. Figure 1 presented the silk glands, cocoons, and microstructures of silks from different species of wild-type silkworms and silk fibroin-deficient mutant silkworms in our research group.

Fluorescent materials with high biocompatibility and bioactivity are widely applied in tracing and labeling [29,35]. When they act as an additive, the fluorescence interferes with some test items. For different practical applications, we selected silk fibroin-deficient mutant silkworms through genetic breeding technology, which could produce sericin with strong fluorescence or weak fluorescence at different wavelengths (Figure 2).

The artificial feed rearing of silkworms is an important developmental direction of silkworm rearing [36]. This silkworm-rearing mode is conducive to the formation of uniform and stable silkworm cocoons through the standardized control of the environment. The uniformity and stability of material properties are very important for their applications in biomedicine [37]. For this reason, anti-diseases silk fibroin-deficient mutant silkworms were employed to cross with silkworms with a high adaptability to artificial feed. We obtained a new silk fibroin-deficient mutant silkworm with an adaptability to artificial feed through successive selection and backcrossing, which could produce uniform cocoons with high cocoon layer ratios (Figure 3).

The quality of silkworm cocoons depends on the rearing and mounting conditions. High-quality silkworm feed (mulberry leaves or artificial feed) was used to rear silkworms under standardized feeding conditions for obtaining high-quality pure sericin cocoons [38]. Considering that the temperature, humidity [39], and airflow (especially the humidity) in the mounting conditions could affect the properties of sericin, we optimized the mounting conditions and realized the standardization of the parameters to maintain the uniformity and stability of silkworm cocoons [40]. In addition, we found that the silk secretion environment of silkworms influences the properties of cocoons during the production process of sericin raw materials. In particular, environmental humidity significantly affected the secondary structure composition of sericin in silkworm cocoons. Based on our research, it was appropriate to control the environmental temperature of silkworm silk production at 23–28 °C, and the humidity at 60–70%. Finally, the sick and injured individuals were first removed to prevent contamination of the cocoons in the cocoon-picking process.

## 3. Characteristics and Advantages of Sericin from Fibroin-Deficient Mutant Silkworms

### 3.1. Extraction Processes of Sericin

In general, extracting sericin from wild silkworm cocoons mainly adopts methods involving high-temperature and high-pressure, urea, sodium carbonate, and acid extraction [41]. However, these relatively rough methods will seriously damage the natural structure of the sericin, leading to the degradation of sericin. In addition, it is difficult to completely remove the chemical agents introduced by the above methods, resulting in serious restrictions of the practical applications of sericin [42].

Based on the characteristics of sericin secreted by mutant silkworms, we established or optimized various extraction methods of sericin and obtained sericin for meeting different needs, which greatly expanded the application of sericin (Figure 4a,b).

The extraction methods of sericin from fibroin-deficient mutant silkworms were as follows:(1)The cocoons were crushed at a low temperature and then sieved to 100 mesh, which could accelerate the hydration of sericin, shorten the time of sericin dissolution, and further reduce the degradation of sericin.(2)Sericin with a low degradation degree could be extracted from cocoons using a relatively mild method of LiBr [16].(3)Low molecular weight sericin was obtained by the high-temperature and high-pressure method or enzymolysis method [43,44].(4)The concentration of sericin reached 16% (*w*/*v*) by extracting sericin from silk glands of silkworms directly at the mature stage in the 5th instar.

### 3.2. Advantages of Sericin from Fibroin-Deficient Mutant Silkworms

Sericin extracted from silk fibroin-deficient mutant silkworm cocoons has many advantages. Firstly, sericin extracted from cocoons using a relatively mild method of LiBr has a low degradation degree. Sericin extracted from cocoons using a relatively mild method of LiBr (Figure 4c, lane 2) was similar to that extracted from the silk glands of silkworms directly [45], and maintained its original protein structure. However, sericin extracted by using boiling water could disrupt the protein structure and produce a low molecular weight, and thus the electrophoretic band had diffusion (Figure 4c, lane 1). Secondly, the solubility of sericin in fibroin-deficient mutant cocoons was much higher than that in wild-type cocoons in hot water. Sericin in mutant cocoons was dissolved completely at 121 °C for 20 min, while only a fraction of wild-type cocoons could be dissolved. It is well known that sericin is comprised of various bioactive peptides [8]. Therefore, the ingredients of sericin isolated from fibroin-deficient cocoons were different from those of wild-type cocoons, even when using the same extraction method. Thirdly, the contents of the free amino group in sericin extracted from mutant cocoons were much higher than that from wild-type cocoons. For example, the content of free amino end groups in the sericin isolated from fibroin-deficient silkworm cocoons (180 Nd-s) was as high as 12 times that of wild-type silkworms cocoons (Jingsong A) (to be published).

## 4. Preparation and Application of Sericin-Based Biomaterials

As a natural macromolecular, sericin from fibroin-deficient mutant silkworms can be readily engineered into various types of biomaterials, such as hydrogels, films, sponges, scaffolds, conduits, and so on [7,46,47]. Hydrogels are a class of three-dimensional (3D) polymer networks which can adjust their physicochemical properties to meet specific needs under different conditions [48]. As early as 2014, an author in our team prepared injectable and fluorescent sericin hydrogels by crosslinking glutaraldehyde with low-degradation sericin extracted with the LiBr method from silk fibroin-deficient mutant cocoons (Figure 5a) [16]. Considering the toxicity of crosslinkers and the poor transparency of the prepared sericin hydrogels, we extracted sericin from the white fibroin-deficient mutant cocoons and used the ultrasonic method to prepare an injectable sericin hydrogel with high transparency, good elasticity, and high biocompatibility (Figure 5b) [49]. The difficulty is in the sterilization of silk-worm cocoons and the extracted sericin protein solution’s failure to pass the filtration membrane (0.22 μm), while the existing sterilization methods of sericin hydrogels can have adverse effects on the material. Therefore, we synthesized a sterile self-assembled sericin hydrogel by using the high-temperature and high-pressure method (Figure 5c) [43]. A high extraction rate of sericin (up to 95%) and a high concentration of sterile sericin solution (up to 15% *w*/*v*) were obtained directly under high-temperature and high-pressure conditions. The gelation time of these kinds of sericin hydrogels could be controlled within a few minutes to hours by adjusting the concentration, pH, and temperature of sericin protein solution. In addition, these hydrogels without toxic crosslinkers formed by self-assembly in a sterile environment can be directly used as biomaterials for cell culture or tissue engineering. In many cases, a key limitation to the application of hydrogels is their relatively poor mechanical strength [50]. For this reason, we tried to isolate non-degradable sericin with a high concentration (16% *w*/*v*) from silk fibroin-deficient mutant silkworms and then prepared a strong sericin hydrogel with high elastic modulus (310 kPa) by cross-linking sericin with H_2_O_2_ [45]. This hydrogel has excellent elasticity, good cytocompatibility, and controlled drug delivery (Figure 5d). In addition, we combined sericin with alginate or polyacrylamide to prepare two kinds of interpenetrating double-network hydrogels with excellent properties, namely sericin-alginate hydrogel (Figure 5e) [17] and sericin-polyacrylamide hydrogel (Figure 5f) [51]. The above sericin-based hydrogels had various properties, including a simple fabrication process, sterility, injectability, high mechanical strength, satisfactory bioactivities, etc.

Except for hydrogels, various biomaterials were exploited based on sericin from fibroin-deficient mutant silkworms, including films, sponge, scaffolds, and conduits. Beneficial for their satisfactory properties of sericin, sericin-based biomaterials are widely applied in various fields. Here, we summarize different types of sericin-based materials and their biomedical applications (Table 1).

### 4.1. Cell Culture

With the rapid development of the biomedicine industry, the scale of cell culture is growing, as well as the market demand for cell culture medium. Fetal bovine serum (FBS), as one of the main components of culture medium, has occupied tens of billions of dollars in the market [58,59]. However, there are many problems in the expansion and application of FBS, such as high prices, the risk of viral infection, animal ethics issues, and insufficient resources. Hence, developing inexpensive, safe, and effective FBS substitutes for cell culture has become an important research topic and a hot issue in cell culture fields [23,60]. In this contribution, our group provided FBS substitutes of sericin protein without toxic and harmful substances by extracting sericin from three silk fibroin-deficient mutant cocoons through high-temperature and high-pressure or enzymolysis methods. We then systematically evaluated the effects of sericin protein and its hydrolysate on culturing several cell lines, including cell adhesion, cell viability, cell growth, and proliferation. The results showed that sericin and its hydrolysate from different varieties of cocoons could greatly promote cell proliferation. Moreover, we compared the function of sericin and FBS on cell culture. The cell viability results showed that sericin protein and its hydrolysate groups were similar to the control (10% FBS), and there was no significant difference in cell morphology in different groups (Figure 6, to be published). In addition, compared with FBS medium, cells cultured in sericin medium showed similar cell morphology, similar or higher cell survival, a lower population doubling time (PDT), and a higher percentage of S phases with a higher G2/G1 ratio, indicating that sericin is beneficial to cell growth and proliferation [23]. Moreover, many studies demonstrated that sericin can promote the proliferation and maintain the functions of various types of cells, including normal mammalian cells, tumor cells, and insect cells [58,59]. For instance, a culture medium containing 0.01% sericin could maintain the secretion function of islet cells, thus exerting the hypoglycemic effects of islet cells [61]. These aforementioned benefits of sericin on cell culture expand its applications (for example, in tissue engineering).

### 4.2. Tissue Engineering

With the development and utilization of sericin from fibroin-deficient silkworms, sericin-based biomaterials have been increasingly investigated and utilized in skin injury repair [25,54,55], bone and cartilage tissue engineering [26,53], and nerve regeneration [21,22,24,30].

Skin injury repair is challenging for traditional wound dressings (such as gauze, bandages, and cotton pads). Natural biomaterials are favored for preparing bioactive wound dressings due to their functions of cell proliferation and differentiation, and good biocompatibility [62]. Sericin fibers with non-destructive structures have excellent physicochemical properties and biological activities, while it is attractive to develop a bioactive wound dressing based on native sericin fiber [55]. Our group first reported designing and performing a flat and uniform natural sericin wound dressing using fibroin-deficient silkworms through the directional genetic breeding technology. The obtained sericin wound dressing had a porous fibrous network structure with high porosity, thus achieving excellent air permeability (Figure 7). Moreover, the sericin wound dressing presented softness, high mechanical strength, and a high water absorption capacity. More importantly, the sericin fiber scaffold had good biocompatibility, which could support cell growth, proliferation, and migration. When tested in a mouse model of full-thickness skin wounds, the sericin wound dressing effectively promoted wound healing [55]. In addition, studies reported that films or hydrogels formed by sericin served as wound dressings for wound healing [54]. In another work, we prepared a highly bioactive silk protein wound dressing (SPD) with a natural silk fiber-sericin hydrogel interpenetrating double network structure. SPD combines the dual characteristics of natural silk protein and sericin hydrogel, which can promote wound healing in mice [25]. New biomaterials prepared via sericin combination with other polymers (such as polyacrylamide) have great potential for wound healing [51].

Currently, the application of sericin from silk fibroin-deficient silkworms in other tissue injury repair areas mainly refers to nerve injury repair, stroke, and myocardial infarction. The excellent biocompatibility of sericin is keystone factor for its application [7]. As mentioned above, Wang’s team first reported a genipin cross-linked chitosan-sericin 3D scaffold for the repair of chronic nerve compression [30]. This mainly benefits from the function of sericin to promote nerve regeneration. Sericin has been used to construct peripheral neural regeneration due to its good biocompatibility and biodegradability [21,22,24]. In addition, the neural regeneration function of a nerve guidance conduit prepared by combining sericin with silicone has been verified [21]. Furthermore, a carbon nanotube (CNT)/sericin nerve conduit with electrical conductivity and suitable mechanical properties was developed and applied for nerve repair. This CNT/sericin nerve conduit possessed electrical conductivity, favorable properties (including physicochemical properties and bioactivities), and suitable mechanical properties. When this conduit was applied in a rat sciatic nerve injury model, the CNT/sericin conduit combined with electrical stimulation effectively promoted both structural repair and functional recovery [57]. It was reported that using sericin fabricated a genipin-cross-linked sericin hydrogel (GSH), which supported the effective attachment and growth of neurons in vitro. Notably, sericin has intrinsic neurotrophic and neuroprotective functions, which endowed GSH as a potential neuronal cell delivery vehicle for ischemic stroke repair [20]. For acute myocardial infarction treatment, sericin hydrogel was injected into the myocardial infarction area and they found that sericin hydrogel could reduce scar formation and the infarct area, increase the wall thickness and neovascularization, inhibit the inflammatory response and apoptosis induced by myocardial infarction, and thus significantly improve cardiac function [31]. Although there are only a few examples which demonstrate that sericin from silk fibroin-deficient silkworms could be used well in disease therapy, it will be widely used in disease therapy in the future with the deepening of research.

In terms of bone tissue engineering, sericin can enhance the adhesion, proliferation, and differentiation of osteoblast cells, thus promoting bone regeneration [63]. Sericin has been functionalized to be sericin methacryloyl (SerMA), which formed an in situ hydrogel under UV light through photo-crosslinking. SerMA hydrogel possesses excellent biocompatibility and was shown to adhere to chondrocytes, and promote the growth and proliferation of chondrocytes even in nutrition-lacking conditions. Surprisingly, the in vivo implantation of SerMA hydrogels loaded with chondrocytes effectively formed artificial cartilages after 8 weeks [26]. Following that, the team mixed graphene oxide into SerMA hydrogel to improve its mechanical strength [53]. The sericin in the hydrogel can promote the migration of bone marrow stromal cells (BMSCs), while the embedded graphene oxide can promote the osteogenic differentiation of BMSCs. Therefore, this hydrogel demonstrated a significant effect on skull regeneration in mice [53].

### 4.3. Drug Delivery

Sericin-based biomaterials, such as hydrogels, films, and sponges, are usually used as carriers for drug delivery [28]. Hydrogels formed by sericin from silk fibroin-deficient silkworms could load drugs or growth factors, exhibiting the function of controlled release [16,17,28]. Zhang et al. proved that increasing the contents of sericin in interpenetrating network (IPN) hydrogels (comprising interwoven sericin and alginate sericin and alginate) can improve its sustained-release ability. This study indicated that sericin provides a convenient way to regulate and control drug release [17]. As aforementioned, our team evaluated the release behavior of the sericin hydrogel formed by sericin from the silk gland of silk fibroin-deficient mutant silkworms cross-linking H_2_O_2_. An antitumor drug, adriamycin hydrochloride (DOX-HCl), was chosen as the model drug, and we observed a sustained drug release within 40 days [45]. Wang et.al fabricated a kind of hydrogel with three-dimensional (3D) networks and subsequently revealed its drug release performance, suggesting that this sericin hydrogel has potential as a carrier for drug delivery in vivo [16]. In another work, a genipin cross-linked chitosan-sericin 3D hydrogel scaffold was designed and prepared for in vivo delivering nerve growth factor (NGF) [30]. In our work, a series of hydrogels were prepared by self-assembly or crosslinking with other polymers [43,45,49,51]. These hydrogels possess a high swelling rate, which is beneficial for drug delivery. In addition to hydrogels, films and sponges formed by sericin from silk fibroin-deficient silkworms have also been used for drug delivery. In Ayumu Nishida’s study, the authors used sericin to prepare films and sponges and then examined the release properties of the charged protein, fluorescein isothiocyanate-albumin (FA). The results demonstrated that FA was sustained-released from film and sponges in vitro. For the in vivo release, FA remained for 3–6 weeks or more in rats. The concentration of sericin is a crucial factor for charged drug effective release [28]. These findings indicated that sericin is an excellent biomaterial for drug delivery.

### 4.4. Cosmetics

Sericin has attracted great attention in the field of cosmetics due to its natural biological activities, including moisturization, anti-oxidation, antibacterial, and anti-inflammatory properties, its inhibition of tyrosinase, and polyphenol oxidase activities [64,65,66]. The antioxidant properties of sericin are closely related to its activities of scavenging reactive oxygen species (ROS), inhibiting lipid peroxidation, and anti-tyrosinase and anti-elastase properties [67,68]. In addition, sericin could enhance the activity of antioxidant enzymes, such as superoxide dismutase (SOD), catalase (CAT), and glutathione peroxidase (GPX) [69]. The antioxidant activity of sericin benefits from its high serine and threonine content [68]. Moreover, it was found that sericin has inhibitory effects on Staphylococcus epidermidis, Staphylococcus aureus, and Bacillus subtilis. The reason is that sericin has a positive amino acid side-chain due to its carboxyl group being protonated under acidic conditions, thus having antibacterial activity [70]. In addition, sericin contains various amino acids with hydrophilic side groups (about 80%), such as serine (30 to 33%), which has a superb water absorption capacity. The sericin may also prevent the loss of water by forming a smooth and soft film on the skin surface [71].

## 5. Conclusions and Outlook

Sericin has excellent biological activity and abundant resources. The presence of mutant varieties of silkworms, especially, have brought great conveniences to our development and application of sericin. Through the optimization of the sericin extraction process, the further study of sericin biological activity, and the exploitation of new sericin-based materials, sericin will expand its application in various fields. In addition to cell culture, tissue engineering, drug delivery, and cosmetics, sericin from silk fibroin-deficient silkworms also has great potential in applications for functional foods. At present, there have been few studies about sericin from silk fibroin-deficient silk-worms in functional food. Sericin consists of 18 amino acids, including serine, aspartic acid, glycine, threonine, tyrosine, alanine, etc. [7]. Among them, glycine and serine can promote glycaemic decomposition and reduce the amount of cholesterol in the blood [72]. Alanine can stimulate the cell vitality of the hepatocyte, thus accelerating alcohol metabolism [73]. And tyrosine has significant effects on the prevention and treatment of Alzheimer’s disease [74]. Therefore, there is a possible trend to develop and utilize sericin from silk fibroin-deficient silkworms as raw materials for commercial functional food. In summary, sericin has promising applications and commercial value in the future.

## Figures and Tables

**Figure 1 polymers-15-02941-f001:**
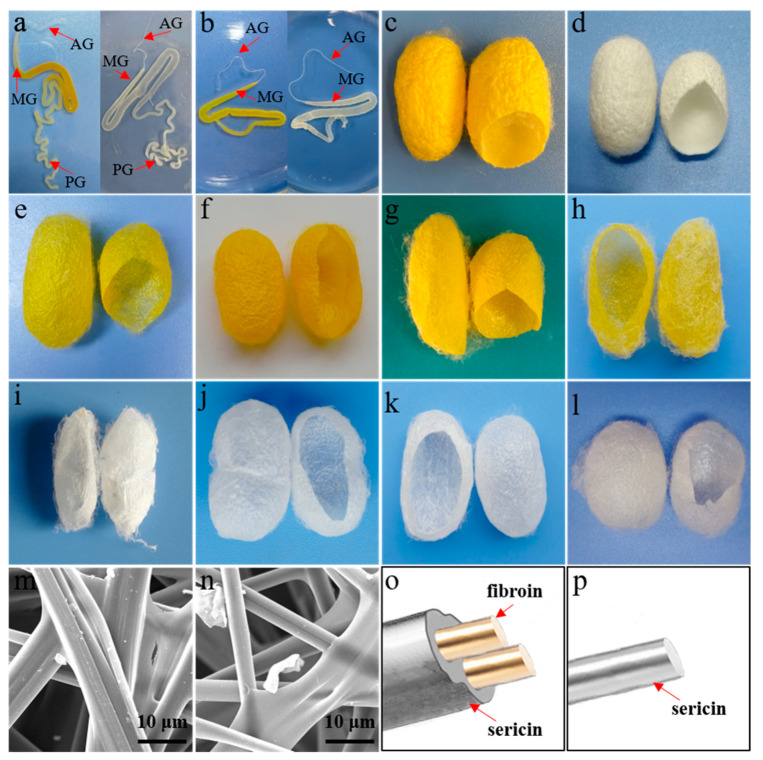
Silk glands, cocoons, and microstructures of silks from wild-type silkworms and silk fibroin-deficient mutant silkworms. (**a**,**b**) The photos of silk glands isolated from wild-type silkworms (**a**) and silk fibroin-deficient mutant silkworms (**b**), respectively. The anterior silk gland (AG), middle silk gland (MG), and posterior silk gland (PG) are labeled using red arrows. There is no AG in the silk glands of silk fibroin-deficient mutant silkworms. (**c**,**d**) The representative photograph of the cocoons produced by wild-type silkworms. (**e**–**l**) The representative photograph of cocoons produced by different varieties of fibroin-deficient silkworms. (**m**,**n**) The microstructures of the wild-type silkworm cocoons (**m**) and silk fibroin-deficient mutant silkworm cocoons (**n**), respectively. (**o**,**p**) The model diagrams of silk fibers secreted from wild-type silkworm (**o**) and fibroin-deficient mutant silkworm (**p**).

**Figure 2 polymers-15-02941-f002:**
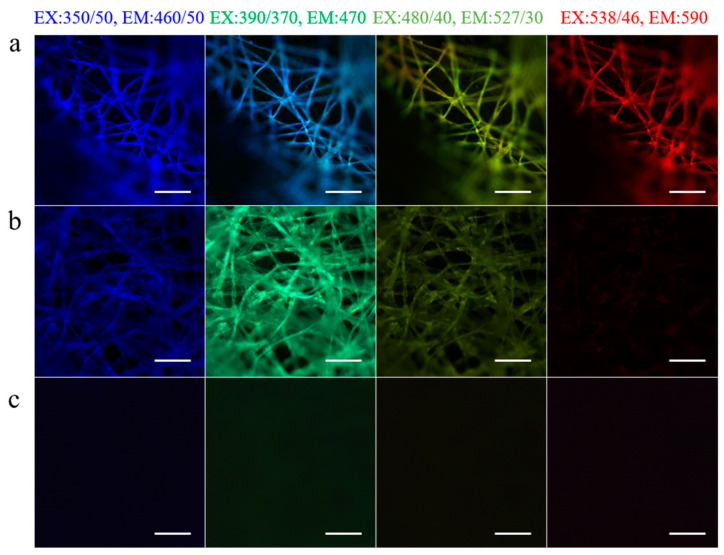
Fluorescence of fibroin-deficient silkworm cocoons. (**a**–**c**) The fluorescence images of the random regions from three varieties of silkworm cocoons were imaged under the light at different wavelengths. Scale bars, 50 μm.

**Figure 3 polymers-15-02941-f003:**
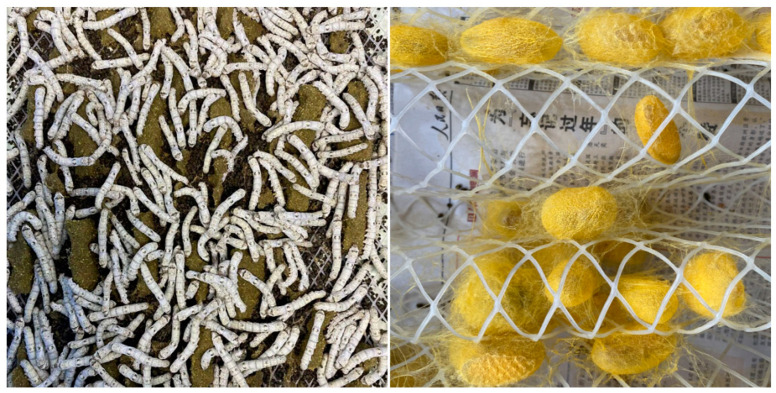
Silkworms with high adaptability to artificial diet and their cocoons produced by raising the artificial diet for all ages.

**Figure 4 polymers-15-02941-f004:**
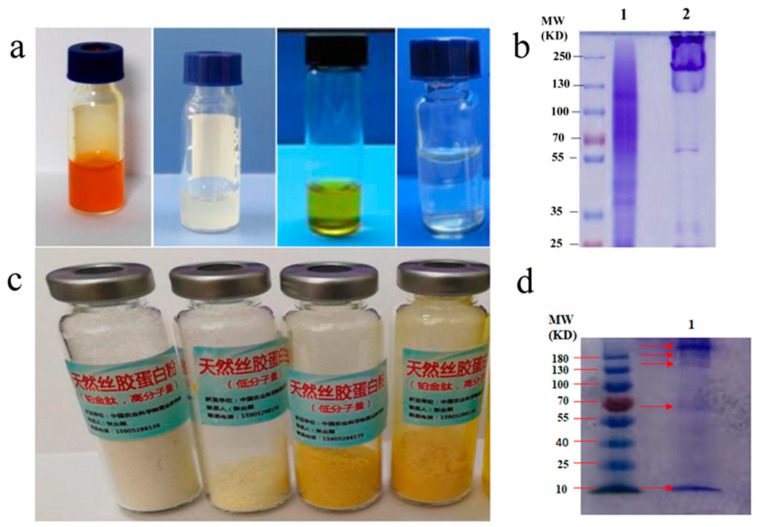
A series of sericin solutions and sericin powders prepared in different ways. (**a**) Left column and middle left are the high concentrations of native sericin solutions isolated from silk gland of different varieties of silk fibroin-deficient silkworms. Middle-right and right column are the sericin solutions with low degradation extracted from different varieties of fibroin-deficient silkworm cocoons using LiBr methods. (**b**) Sericin protein powders prepared from fibroin-deficient cocoons in different ways. From left to right: white high molecular sericin powder, white low molecular sericin powder, yellow low molecular sericin powder, yellow low molecular sericin powder, yellow low molecular sericin powder. (**c**) The protein profiles of the sericin solution extracted from fibroin-deficient silkworm cocoons by using boiling water (**lane 1**) and 6 M LiBr at 35 °C (**lane 2**). (**d**) The protein profiles of the sericin solution extracted from the silk gland of fibroin-deficient silkworms (**lane 1**).

**Figure 5 polymers-15-02941-f005:**
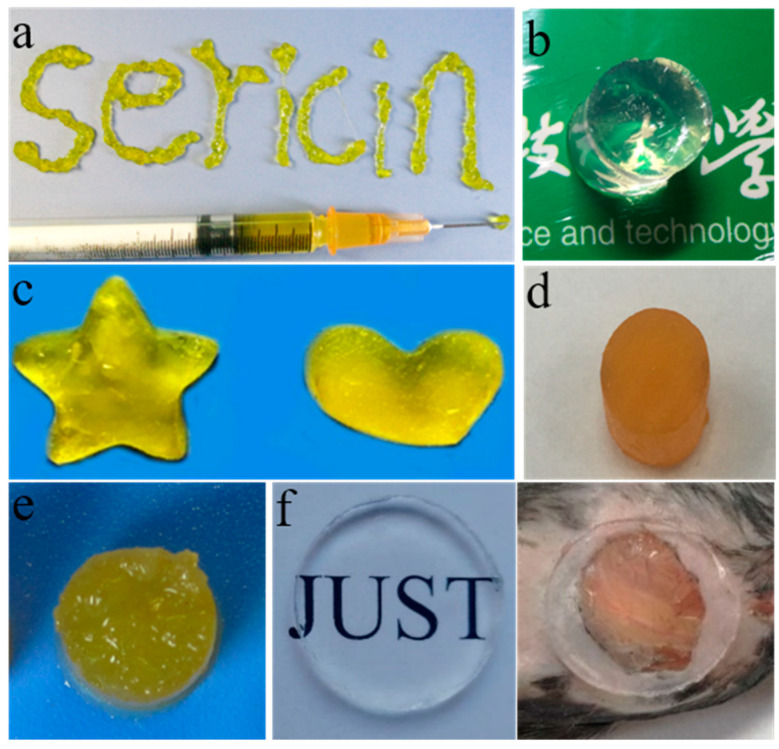
Hydrogels fabricated by sericin isolated from fibroin-deficient silkworm cocoons or silk glands of silkworms. (**a**) An injectable, photoluminescent, cell-adhesive 3D hydrogel formed by covalent crosslinking [16]. (**b**) A highly transparent, elastic, injectable sericin hydrogel induced by ultrasound [49]. (**c**) A sterile self-assembled sericin hydrogel [43]. (**d**) A robust sericin hydrogel [45]. (**e**) A sericin-alginate interpenetrating network hydrogel [17]. (**f**) A transparent sericin-polyacrylamide interpenetrating network hydrogel [51].

**Figure 6 polymers-15-02941-f006:**
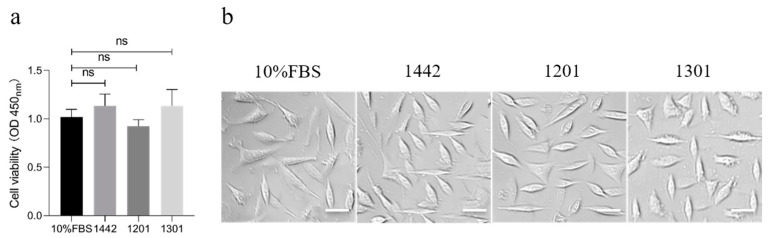
The cell viabilities (**a**) and morphologies (**b**) of NIH3T3 cells cultured using high sugar DMEM medium containing 10% fetal bovine serum (FBS) or 0.2% (*w*/*v*) sericin isolated from cocoons of 3 silkworm varieties (1201, 1301, and 1442). ns in figure a representative no significant difference.

**Figure 7 polymers-15-02941-f007:**
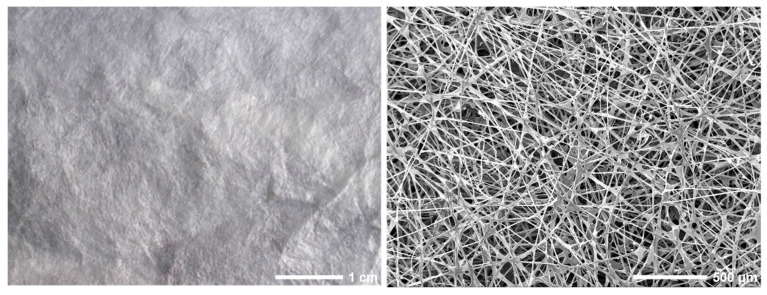
Sericin wound dressing (**left column**) and its microstructure (**right column**) [55].

**Table 1 polymers-15-02941-t001:** Biomedical applications of sericin from fibroin-deficient silkworms.

Materials	Applications	Reference
**Hydrogels**		
Sericin	Tissue engineering	[43,45,49]
Sericin	Wound dressing	[25]
Sericin	Ischemic Stroke	[20]
Sericin	Ischemic myocardial infarction	[31]
Sericin	Cartilage regeneration	[26]
Sericin/Methacrylate	Skin wound healing	[52]
Sericin/Graphene oxide	Calvarial bone regeneration	[53]
Sericin/Alginate	Cell culture and drug delivery	[17]
Sericin/Glutaraldehyde	Cell culture and drug delivery	[16]
Sericin/Polyacrylamide	Visualized dressing	[51]
**Films**		
Sericin	Wound dressing	[54]
Sericin	Drug delivery	[28]
**Sponges**		
Sericin	Drug delivery	[28]
**Scaffolds**		
Sericin	Wound dressing	[55]
Sericin/Chitosan	Chronic nerve compression treatment	[30]
Sericin/Carbon-Nanotubes (CNTs)	Ischemic stroke damage treatment	[56]
**Conduits**		
Sericin	Peripheral nerve regeneration	[22]
Sericin/Silicone	Peripheral nerve regeneration	[21]
Sericin/Clobetasol	Peripheral nerve regeneration	[24]
Sericin/CNTs	Peripheral nerve regeneration	[57]

## Data Availability

No new data were created or analyzed in this study.

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
