# Peer review of "Sericin from Fibroin-Deficient Silkworms Served as a Promising Resource for Biomedicine"

_polymers, 2023, doi:10.3390/polym15132941_

Round 1

Reviewer 1 Report

I this manuscript “Sericin from fibroin-deficient silkworms served as a promising resource for biomedicine” the authors presented an overview of silkworm varieties resources, especially silk fibroin-deficient mutant silkworms. They optimized various extraction methods of sericin and summarized the characteristics and advantages of sericin. Finally, they developed a series of sericin-based biomaterials for promising applications to a diverse set of needs. This review, of the authors in this area is interesting, some observations to the document I present below:

1)      Both in the abstract and in the conclusions, it is indicated the importance applied in various fields, such as biomedicine, cell culture, cosmetics, and functional food. Although details are given in the introduction, it would be good for the reader to know some examples from the beginning.

2)      Throughout the document the first name of the plural "we" is used, we understand the great contribution of your group in the area, however, for this type of publication it is not recommended to use it, but it is the preference of the authors.

3)      Please give the date of the statistical information provided on line 60.

4)      Figures in their description (caption) the authors provide sufficient information, which is correct, but they are not used much in the discussion.

Author Response

Reviewer #1. In this manuscript “Sericin from fibroin-deficient silkworms served as a promising resource for biomedicine” the authors presented an overview of silkworm varieties resources, especially silk fibroin-deficient mutant silkworms. They optimized various extraction methods of sericin and summarized the characteristics and advantages of sericin. Finally, they developed a series of sericin-based biomaterials for promising applications to a diverse set of needs. This review, of the authors in this area is interesting, some observations to the document I present below:

Comment 1:  Both in the abstract and in the conclusions, it is indicated the importance applied in various fields, such as biomedicine, cell culture, cosmetics, and functional food. Although details are given in the introduction, it would be good for the reader to know some examples from the beginning.

Response 1: Thanks for the reviewer’s kind and insightful advice. According to the adjusted contents of the review, we have added some examples for discussing the application of sericin in cell culture, tissue engineering, drug delivery, and cosmetics in the revised manuscript.

Comment 2:  Throughout the document the first name of the plural "we" is used, we understand the great contribution of your group in the area, however, for this type of publication it is not recommended to use it, but it is the preference of the authors.

Response 2: Thanks for your kind suggestions. We have made some changes to this writing type in the revised manuscript.

Comment 3:   Please give the date of the statistical information provided on line 60.

Response 3: Thanks for this kind advice. Currently, the annual output of fresh silkworm cocoons is about one million tons in the world, of which cocoon shells account for 20%. Sericin glued together silk fibroin to form cocoons shells, which accounts for about 25%. Therefore, the annual output of sericin is calculated to be 50,000 tons. We have added the relevant reference in the revised manuscript.

Comment 4:  Figures in their description (caption) the authors provide sufficient information, which is correct, but they are not used much in the discussion.

Response 4: This is a good suggestion. We have added new information for figures in the revised manuscript.

Reviewer 2 Report

Minor editing of English language required

Author Response

Reviewer #2. In this review article, the authors presented an overview of our silkworm varieties resources, especially silk fibroin-deficient mutant silkworms and the possible application in biomedical fields. The review article is well written and represents interesting findings. However, it would be better if the authors provide more details regarding

  • Sericin extraction methods

2- Applications of sericin-based systems in drug delivery

Response: Thanks for your kind and insightful advice. As suggested, we have added the sericin extraction methods and applications of sericin-based biomaterials in drug delivery in the revised manuscript.

Minor points

Comment 1: Lines 187-188: The authors mentioned that “sericin 187protein and its hydrolysate groups were superior to the control (10% FBS)….” While in figure 5a no significant differences were observed. Please modify your description.

Response 1:  We are sorry for our carelessness. We have corrected the description in the revised manuscript. “The results of cell viability showed that sericin protein and its hydrolysate groups were similar to the control (10% FBS)….”

Comment 2: Lines 259-260: “This hydrogel has excellent elasticity, good cytocompatibility, and controlled drug delivery (Fig. 8d)”. Please provide more details about controlled drug delivery.

Response 2: Thanks for your kind advice. As suggested, we have added more details about controlled drug delivery in “4.3 Drug delivery” in the revised manuscript.

Our team evaluated the release behavior of the sericin hydrogel formed by sericin from silk gland of silk fibroin-deficient mutant silkworms cross-linking H2O2. An antitumor drug, adriamycin hydrochloride (DOX-HCl) was chosen as a model drug. And we observed a sustained drug release within 40 days (Fibers Polym. 2022, 23, 1826-1833.).”

Reviewer 3 Report

This review is interesting and provides valuable information on sericin from fibroin-deficient silkworms and its potential applications in biomedicines. However, some concerns should be addressed as follows:

1. The authors should highlight the purification approaches of sericin from coccons.

2. The authors should mention the source of all figures presented in this review (Reference Source or permission).

3. The activity of sericin as antioxidant, antibacterial, anti-inflammatory and other biological properties should be discussed and from where they stem.

4. The authors should design a table to summarize the sericin-based biomaterials and their applications.

5. The authors should design a table to summarize the sericin-based biomaterials, including hydrogel, nanofibers, nanopartciles, films and their activities as wound dressings. Additionally, figures should be incorporated to support this section.

6. What about other biomedical implementations of sericin? You should discuss other biomedical applications in brief or modify the title to focus on wound dressing after addressing the above comments.

7. Some update references should be added to support the manuscript; for instance, https://doi.org/10.1016/j.jconrel.2022.11.019; https://doi.org/10.1016/j.ijpharm.2022.122328; https://doi.org/10.1016/j.biomaterials.2022.121638; https://doi.org/10.1186/s12951-021-00774-y).

Some minor mistakes should be addressed.

Author Response

Reviewer #3. This review is interesting and provides valuable information on sericin from fibroin-deficient silkworms and its potential applications in biomedicines. However, some concerns should be addressed as follows:

Comment 1: The authors should highlight the purification approaches of sericin from cocoons.

Response 1: Thanks for your kind advice. We have highlighted the purification approaches of sericin from cocoons in the revised manuscript.

Comment 2: The authors should mention the source of all figures presented in this review (Reference Source or permission).

Response 2: Thanks for your kind suggestions. We have added the relevant references for all figures in the revised manuscript.

Comment 3: The activity of sericin as antioxidant, antibacterial, anti-inflammatory and other biological properties should be discussed and from where they stem.

Response 3: Thanks for the reviewer’s kind and insightful advice. We have added the discussion on the activity of sericin as antioxidant, antibacterial, and other biological properties in section 4.4 of the revised manuscript.

Comment 4: The authors should design a table to summarize the sericin-based biomaterials and their applications.

Response 4: Thanks for this kind advice. We have added a table to summarize the sericin-based biomaterials, including hydrogels, films, sponges, scaffolds, conduits, and their applications (Table 1) in the revised manuscript.

Comment 5: The authors should design a table to summarize the sericin-based biomaterials, including hydrogel, nanofibers, nanopartciles, films and their activities as wound dressings. Additionally, figures should be incorporated to support this section.

Response 5: Because there are many comprehensive reviews about sericin (including sericin from fibroin-deficient silkworms and wild-type silkworms) (Biomaterials. 2022 Jun 17;287:121638.; Journal of Controlled Release 353 (2023) 303–316), we here focused on the application of sericin from fibroin-deficient silkworms. In the revised manuscript, we summarized the application of sericin-based biomaterials (including hydrogels, films, sponges, scaffolds, conduits, and their applications) in the fields of cell culture, tissue engineering, drug delivery, and cosmetics.

Comment 6: What about other biomedical implementations of sericin? You should discuss other biomedical applications in brief or modify the title to focus on wound dressing after addressing the above comments.

Response 6: Thanks for the reviewer’s kind and insightful advice. We have added the discussion about other biomedical applications, including cell culture, tissue engineering, drug delivery, and cosmetics in section 4, and the potential application of functional food in section 5.

Comment 7: Some update references should be added to support the manuscript; for instance, https://doi.org/10.1016/j.jconrel.2022.11.019; https://doi.org/10.1016/j.ijpharm.2022.122328; https://doi.org/10.1016/j.biomaterials.2022.121638; https://doi.org/10.1186/s12951-021-00774-y).

Response 7: Thanks for the reviewer’s kind advice. We have added these references and other references to support the manuscript.

Reviewer 4 Report

The present review covers some general aspects and applications of sericin from fibroin-deficient silkworms. This topic is really interest and the authors are expert in the field, since they produced these silk-fibroin deficient mutant silkworms.

However, the manuscript in some parts, it sounds too much as a promotion of the authors work in the field or a sponsoring of their products. This can have an impact in the general scientific soundness of the manuscript. I suggest revising all the manuscript to achieve a more rigorous scientific style and content.

Data presented in Figure 1, Figure 2, Figure 3, Figure 4, Figure 5, Figure 8 are not referenced. From which work are they reproduced?

The production process of the mutant silkworm and the advantages of sericin from fibroin-deficient silkworms in comparison to that of wild-type silkworms must be better explained (Paragraph 2 and 3 must be implemented).

Lines 151-152 and Lines 202-203 and Figure 6 are not appropriated for a scientific manuscript.

In caption of Figure 4, it is not correctly described the difference between Figure 4C and Figure 4D (electrophoresis runs).

For all presented applications, it lacks a deep discussion about the potential of the proposed “sericin from fibroin-deficient silkworms” with sericin from wild type silkworms.  Paragraph 4 must be implemented

Conclusions must be expanded and revised since are too much generic.

Author Response

Reviewer #4.  The present review covers some general aspects and applications of sericin from fibroin-deficient silkworms. This topic is really interest and the authors are expert in the field, since they produced these silk-fibroin deficient mutant silkworms.

Comment 1: However, the manuscript in some parts, it sounds too much as a promotion of the authors work in the field or a sponsoring of their products. This can have an impact in the general scientific soundness of the manuscript. I suggest revising all the manuscript to achieve a more rigorous scientific style and content.

Response 1: Thanks for the reviewer’s kind advice. We have revised the manuscript to achieve a more rigorous scientific style and content.

Comment 2: Data presented in Figure 1, Figure 2, Figure 3, Figure 4, Figure 5, Figure 8 are not referenced. From which work are they reproduced?

Response 2: Thanks for the kind suggestions. We have added the relevant references for all figures in the revised manuscript.

Comment 3: The production process of the mutant silkworm and the advantages of sericin from fibroin-deficient silkworms in comparison to that of wild-type silkworms must be better explained (Paragraph 2 and 3 must be implemented).

Response 3: This is a good suggestion. We have recomposed and summarized extraction processes and the advantages of sericin from fibroin-deficient silkworms in comparison to that of wild-type silkworms in section 3 of the revised manuscript.

Comment 4: Lines 151-152 and Lines 202-203 and Figure 6 are not appropriated for a scientific manuscript.

Response 4: Thanks for the kind suggestions. We have removed these contents from the original manuscript.

Comment 5: In the caption of Figure 4, it is not correctly described the difference between Figure 4C and Figure 4D (electrophoresis runs).

Response 5: Thanks for the kind suggestions. We have corrected the description in the revised manuscript. “Sericin extracted from cocoons by a relatively mild method of LiBr (Fig. 4c, lane 2) was similar to that from silk glands of silkworms directly (Fig. 4d), which maintained its original protein structure. However, sericin extracted by using boiling water disrupts the protein structure to obtain low molecular weight, thus the electrophoretic band was diffusion (Fig. 4c, lane 1).”

Comment 6: For all presented applications, it lacks a deep discussion about the potential of the proposed “sericin from fibroin-deficient silkworms” with sericin from wild type silkworms.  Paragraph 4 must be implemented

Response 6: Thanks for the reviewer’s kind and insightful advice. There are many comprehensive reviews about sericin (including sericin from fibroin-deficient silkworms and wild-type silkworms) (Biomaterials. 2022 Jun 17;287:121638.; Journal of Controlled Release 353 (2023) 303–316), thus we here focused on the application of sericin from fibroin-deficient silkworms. And we have added a deep discussion about the potential application of sericin-based biomaterials in the fields of cell culture, tissue engineering, drug delivery, and cosmetics in section 4 of the revised manuscript.

Comment 7: Conclusions must be expanded and revised since are too much generic.

Response 7: Thanks for your kind advice. We have expanded the conclusion in the revised manuscript.

Round 2

Reviewer 3 Report

The authors addressed the major comments carefully.

Author Response

Thank you for your positive comments

Reviewer 4 Report

There are no still no references for Figure 1, Figure 2 and Figure 3. Are they original or reproduced from other articles?

Some revisions in English language are required.

Author Response

 Yes, Figure 1, Figure 2 and Figure 3 are  all original.